

# A genetic algorithm-based energy-aware multi-hop clustering scheme for heterogeneous wireless sensor networks

R. Muthukkumar[1], Lalit Garg[2], K. Maharajan[3], M. Jayalakshmi[3], Nz Jhanjhi[4], S. Parthiban[5] and G. Saritha[6]

[1] Department of Information Technology, National Engineering College, Kovilpatti, Thoothukudi, Tamil Nadu, India
[2] Department of Computer information Systems, Faculty of Information and Communication Technology, University of Malta, Msida, Malta, Malta
[3] Department of Computer Science and Engineering, School of Computing, Kalasalingam Academy of Research and Education, Krishnankoil, India
[4] School of Computer Science, Taylor's University, Subang Jaya, Selangor, Malaysia
[5] Saveetha School of Engineering, Saveetha Institute of Medical and Technical Sciences, Chennai, Tamilnadu, India
[6] Sri Sairam Institute of Technology,, Chennai, Tamilnadu, India

## ABSTRACT

**Background:** The energy-constrained heterogeneous nodes are the most challenging wireless sensor networks (WSNs) for developing energy-aware clustering schemes. Although various clustering approaches are proven to minimise energy consumption and delay and extend the network lifetime by selecting optimum cluster heads (CHs), it is still a crucial challenge.

**Methods:** This article proposes a genetic algorithm-based energy-aware multi-hop clustering (GA-EMC) scheme for heterogeneous WSNs (HWSNs). In HWSNs, all the nodes have varying initial energy and typically have an energy consumption restriction. A genetic algorithm determines the optimal CHs and their positions in the network. The fitness of chromosomes is calculated in terms of distance, optimal CHs, and the node's residual energy. Multi-hop communication improves energy efficiency in HWSNs. The areas near the sink are deployed with more supernodes far away from the sink to solve the hot spot problem in WSNs near the sink node.

**Results:** Simulation results proclaim that the GA-EMC scheme achieves a more extended network lifetime network stability and minimises delay than existing approaches in heterogeneous nature.

# INTRODUCTION

The latest technology development in wireless communication, sensing devices, and microelectronics have opened new frontiers in wireless sensor networks (WSNs). Critical WSNs applications include environmental monitoring (*Lanzolla & Spadavecchia, 2021*;

Corresponding authors
Lalit Garg, lalit.garg@um.edu.mt
Nz Jhanjhi,
NoorZaman.Jhanjhi@taylors.edu.my

*Sampathkumar et al., 2020*), transport (*Sant et al., 2021*; *Kumar et al., 2020*; *Andrushia et al., 2021*), surveillance systems (*Garg et al., 2020a*; *Chukwu, Garg & Zahra, 2021*), healthcare (*Chakraborty et al., 2020b, 2020a*; *Agrawal et al., 2019*; *Bonello et al., 2018*; *Audu et al., 2017b, 2017a*; *Bugeja, Garg & Audu, 2016*; *Bugeja & Garg, 2015*; *Agrawal, Garg & Dauwels, 2013*), emotions recognition and monitoring (*Salankar, Mishra & Garg, 2021*; *Jayalakshmi et al., 2021*; *Bhattacharyya et al., 2021*; *Mangion et al., 2020*), home automation (*Garg et al., 2020b*), battlefield monitoring, and industrial automation and control (*Smaragdakis, Matta & Bestavros, 2004*). WSNs contain more sensor nodes capable of sensing the physical phenomenon, packet forwarding, and communicating the packets to the destination. However, each sensor node has limitations, such as limited memory, restricted processing capability, short-range transmission, finite energy resources, and low storage capability. WSNs are becoming a good network for protecting, controlling, and facilitating real-time applications (*Qing, Zhu & Wang, 2006*). The primary constraint of the WSNs is the limited, non-rechargeable battery-powered sensor nodes, and these nodes have used their energy for sensing, sending, and receiving the data. When the sensor battery is drained, several areas in the sensor field will lack coverage, and valuable data from these areas will not reach the sink. Using energy among the nodes and prolonging the network's lifetime are considered the primary challenge for HWSNs.

Various routing and clustering algorithms have been addressed to minimise average energy consumption in WSNs (*Akyildiz et al., 2002*; *Tyagi & Kumar, 2013*; *Heinzelman, Chandrakasan & Balakrishnan, 2000*; *Younis & Fahmy, 2004*; *Shang, 2013*; *Yu et al., 2012*; *Amgoth & Jana, 2015*; *Jin et al., 2013*; *Mahajan, Malhotra & Sharma, 2014*; *Kumar, Aseri & Patel, 2009*). CHs in WSNs allocate the energy to their member nodes to maintain the load-balancing of a cluster (*Saini & Sharma, 2010*). This protocol utilises local coordination to enhance scalability and reduce the large number of data packets communicated to the sink. The clustering protocol splits up the network structure into many clusters, and every cluster contains CH and cluster members (CMs). The CM node collects and sends information to the respective CH in each cluster. Each CH gathers the data packets from the CMs and carries out the data aggregation process. Eventually, the aggregated data is communicated to the sink. The protocol in *Capone et al. (2019)* selects CHs periodically using residual energy and the degree of the nodes. This approach has a low overhead and performs CH selection uniformly across the network. In *Wei et al. (2011)*, the proposed algorithm analyses the routing algorithms in WSNs to extend the network lifetime, save power, and maintain load-balancing. In *Tanwar, Kumar & Rodrigues (2015)*, the energy consumption and nonuniform node distribution in WSNs have been discussed. An energy-aware routing algorithm is presented for cluster-based WSNs to stabilise the energy consumption of the CHs (*Shang, 2013*). Quality of service (QoS) and energy-efficiency requirements are satisfied in practical WSNs (*Yu et al., 2012*). *Amgoth & Jana (2015)* highlights the optimal CH selection cluster formation and maintains the load-balanced network. For a given distance, energy efficiency improves if the data is transmitted through multi-hop. By properly selecting the CHs and the next-hop nodes for multi-hop routing, energy spent by the sensor nodes can be reduced.

*Jin et al. (2013)* analysed the impact of heterogeneity in WSNs, energy level, and hierarchical cluster structures. *Smaragdakis, Matta & Bestavros (2004)* proposed a protocol that prolongs the stability period of sensor nodes in heterogeneous WSNs (HWSNs). In *Qing, Zhu & Wang (2006)*, *Saini & Sharma (2010)*, average network energy and the nodes' residual energy select the optimal CHs. In *Capone et al. (2019)*, the proposed algorithm minimises delay based on signal-to-interference-and-noise-ratio (SINR) in WSNs. *Wei et al. (2011)* found optimal cluster sizes based on the hop count to the sink node. It is also used to extend the network lifetime and minimise energy consumption. Several heterogeneous routing protocols in WSNs (*Tanwar, Kumar & Rodrigues, 2015*) are reviewed and analysed with performance metrics. The algorithm in *Bandyopadhyay & Coyle (2003)* organises the nodes into several clusters in WSNs and generates a hierarchy of CHs. A genetic algorithm (GA) is a meta-heuristic algorithm used to solve optimisation problems (*Pantazis, Nikolidakis & Vergados, 2013*; *Pal & Saraswat, 2017*). GA is an appropriate scheme for solving any clustering problems in WSNs. It is also used to resolve persistent optimisation problems (*Mehta & Pal, 2017*). In this article, HWSNs use GA for solving the multi-hop clustering based on the newly defined fitness function (*Kachitvichyanukul, 2012*).

Existing solutions have the advantage of cluster formation done through the residual energy and prolonging the lifetime of WSNs. However, re-clustering consumes more energy while the end-to-end delay is not minimised. This motivates us to devise an approach for designing energy-aware multi-hop clustering for HWSNs. WSNs with heterogeneous nodes result in better network stability and extend the network lifetime. Energy consumption has been minimised using GA by selecting the optimal CHs during the re-clustering. The main contributions of this article are specified as follows:

- A GA-based energy-aware multi-hop clustering algorithm (GA-EMC) is proposed for selecting the optimal number of CHs dynamically during the re-clustering.
- A framework for optimised transmission scheduling and routing is formulated to reduce the delay under the SINR model for HWSNs.
- A combination of weak and robust sensor nodes using their residual energy mitigates the re-clustering issues.
- For optimising cluster construction, the GA maintains the stability of the nodes in a network.
- A dynamic power allocation scheme for sensor nodes is proposed to have a guaranteed QoS for nodes.

The structure of this article starts with the introduction related to the wireless sensor network, genetic algorithm, and multi-hop clustering paradigms. The following section describes the existing multi-hop clustering algorithms and their issues. In the next section, we present the GA-EMC algorithm, followed by the section which addresses the experimental results and analyses the performance of GA-EMC. The following section is a discussion, and finally, the last section discusses the conclusion.

## RELATED WORKS

This section presents the various modern and advanced multi-hop clustering schemes in WSNs. Many researchers have done some work in multi-hop clustering algorithms based on GA, and an overview of that work is given here. *Saini & Sharma (2010)* developed the probability-based CH selection and decreased the average CHs energy consumption. In *Capone et al. (2019)*, *Wei et al. (2011)*, the spatial distribution of CHs in WSNs by constructing a multi-hop table. It is also used to decrease the CHs when directly transmitted to the sink or base station (BS). The algorithm of *Tanwar, Kumar & Rodrigues (2015)* selects the CHs with higher residual energy and achieves better load-balancing among CHs. In *Bandyopadhyay & Coyle (2003)*, CHs energy consumption was minimised during the data routing process and achieved better time complexity. The procotol of *Pantazis, Nikolidakis & Vergados (2013)* satisfies the QoS requirements in WSNs, and (*Amgoth & Jana, 2015*) addresses the cluster formation and CH selection using weight metrics in HWSNs.

In general, sensor networks can be heterogeneous regarding the initial energy, computational ability of the WSN nodes, and the bandwidth of the links (*Jin et al., 2013*). Designing WSNs with heterogeneous nodes increases the reliability and network lifetime. Computational and link heterogeneity reduces the latency in data transmission (*Smaragdakis, Matta & Bestavros, 2004*; *Qing, Zhu & Wang, 2006*). Various parameters are used to classify the nodes in HWSNs (*Saini & Sharma, 2010*). *Capone et al. (2019)* studied transmission scheduling and multi-hop routing to minimise delay using SINR. The initial energy varies according to the node's distance from the sink to overcome the energy hole problem in multi-hop networks (*Wei et al., 2011*). *Tanwar, Kumar & Rodrigues (2015)* categorised several heterogeneous routing protocols with predefined parameters by enhancing network lifetime and node heterogeneity in WSNs.

GA has been used for the CHs' optimal selection in recent research. The main focus of the GA-based clustering algorithms is the fitness function. The fitness function determines the goodness of an individual to be selected for the next generation (*Bandyopadhyay & Coyle, 2003*). *Pantazis, Nikolidakis & Vergados (2013)* critically analysed the energy-efficient routing protocols for WSNs. The method in *Pal & Saraswat (2017)* is based on biogeography-based optimisation in HWSNs. The fitness value is modified further by incorporating the residual energy of the remaining nodes that enhances the performance. It prolongs the network lifetime (*Mehta & Pal, 2017*). Meta-heuristics techniques are widely applied to solve several clustering problems in WSNs (*Kachitvichyanukul, 2012*; *Bhushan, Pal & Antoshchuk, 2018*). *Fanian & Rafsanjani (2019)* reviewed the various protocols and their properties in WSNs. *Afsar, Mohammad & Tayarani (2014)* investigated and presented more clustering approaches.

*Bari, Jaekel & Bandyopadhyay*'s *(2008)* approach formulates clusters and considers the relay nodes as CHs in two-tiered sensor networks that prolong the relay node lifetime. The method in *Younis, Youssef & Arisha (2003)* extends the network lifetime dynamic route selection and reduces energy consumption. *Liu & Lin (2005)* critically investigated and addressed the power-conserving issues in WSNs, and the algorithm in *Zhang et al. (2017)*

solves the energy balance problem in WSNs. *Gupta & Pandey (2016)* have considered the location of BS and residual energy as clustering parameters to solve an energy hole problem in HWSNs. *Darabkh, Zomot & Al-qudah*'s *(2019)* scheme minimises the average energy consumption and prolongs the lifetime of WSNs.

*Javaid et al.*'s *(2013a)* technique for HWSNs dynamically elects the CH. It extends the network lifetime. *Pal et al. (2015a)* analyses the heterogeneous node locations and selects optimal CH based on the distance between the clusters. The algorithm in *Sarkar & Senthil Murugan (2019)* improves the energy and lifetime of both nodes and networks by choosing the optimal CHs. *Kumar & Kumar (2016)*, *Mann & Singh (2017)* maximise the network energy and extend nodes' network lifetime by selecting optimal CH in WSNs. *Fan*'s *(2013)* method investigates several issues such as energy consumption, coverage, and data routing in WSNs. This method improves the coverage ratio and prolongs network lifetime. *Javaid et al.*'s *(2013b)* scheme increases the node stability period and sends more packets to BS. *Singh & Lobiyal (2012a)* designs an energy-aware cluster by selecting optimal CHs in WSNs.

*Ali, Shahzad & Khan*'s *(2012)* algorithm optimises the clusters in a network and minimises the data traffic and energy dissipation among nodes. *Rakhee & Srinivas (2016)* continuously monitors patients' data by selecting an optimal path in the body area network. It also enhances network lifetime, load balancing, and energy on the overall network. *Pal et al.*'s *(2015b)* method achieves a load-balanced network. It prolongs the lifetime of WSNs by optimising CH selection approach (*Hoang et al., 2014*) that reduces the distance between the CH and CMs in WSNs to improve energy conservation. *Lin et al.*'s *(2012)* approach maximises the lifetime of heterogeneous nodes based on sensing coverage and network connectivity. The approach in *Singh & Lobiyal (2012b)* selects energy-aware clusters and optimal CH based on hop count and locations. *Pal et al. (2020)* considers the GA parameters for enhancing the CH performance in WSNs. *Mhemed et al.*'s *(2012)* approach investigates the cluster formation that reduces energy consumption. *Haseeb et al.*'s *(2019)* method increases energy efficiency and data security against malicious activities. *Zhang et al. (2019)*, *Abo-Zahhad et al.*'s *(2014)* algorithms prolong the network lifetime by selecting the optimal CHs and reducing average energy consumption.

*Delavar & Baradaran (2012)* algorithm reduced energy consumption by selecting chromosomes in different states. *Bayrakli & Erdogan (2012)*, *Liu & Ravishankar (2011)*, *Yu et al. (2011)*, *Zhou et al. (2010)* studied the optimal selection of clusters to extend the WSNs' lifetime. *Bencan et al. (2013)* analyses the spatial distribution of heterogeneous nodes in WSNs and effectively avoids the energy hole problem. *Jin et al.*'s *(2005)* algorithm enhances the reported sensitivity of the nodes and optimises the solution quality in HWSNs management. *Attea & Khalil (2012)* compares the various evolutionary algorithms with network lifetime, node stability period, and energy efficiency. *Elhoseny et al.*'s *(2015)* method optimises heterogeneous sensor node clustering. It dramatically extends the network lifetime. *Huang et al.*'s *(2021)* method was used to minimise the delay and collision and reduce the energy consumption in WSNs. The protocol in *Deng et al. (2021)* improves energy utilisation and minimises the delay in sensor networks. *Chauhan & Soni (2021)* minimises the energy holes and prolongs the network's lifetime in WSNs. In this approach, the network is divided into unequal clusters, and it considers the node's

residual energy and distance to the base station for cluster formation in WSNs. Nodes with the highest energy are considered CH for WSNs (*Toor & Jain, 2019*). *Kamil, Naji & Turki*'s *(2020)* techniques change the WSNs' sink node position dynamically to increase residual energy and prolong the network lifetime.

The method in *Prabaharan & Jayashri (2019)* selects the smart CH in WSNs to prolong the network lifetime. Optimal CH selection is performed for extending the network lifetime of WSN by using various attributes of sensor nodes (*Kalaimani, Zah & Vashist, 2020*; *Rajpoot & Dwivedi, 2019*). *Kashyap et al.*'s *(2019)* algorithm performs load-balancing among sensor nodes for WSNs. Also, it balances the optimal number of CHs and evenly distributes the load among nodes. The technique of *Zhang et al. (2016)* performs that the sensors are scheduled into several disjoint complete cover sets in WSNs and activates them in batch for energy conservation. The algorithm in *Chang et al. (2018)* is suitable for small-scale WSNs and suffers from high network latency due to multiple forwarding operations. *Chang et al. (2019)*, *Wang et al. (2018b)*, *Wang et al. (2020)*, *Wang et al. (2018a)*, *Wang et al. (2022)*, *Tabatabaei (2020)* and *Vijayalakshmi & Anandan (2020)* discussed the various energy management challenges for WSNs.

Many research proposals exist in the related works addressing the energy-efficient hierarchical clustering issues, but node heterogeneity of WSN nodes has not been exploited to its full potential.

Energy efficiency is the essential component in extending the life of WSN systems that are resource-constrained, particularly in terms of energy. The energy-aware clustering algorithms become a significant factor in WSNs since multi-hop clustering methods relate to the network's communication operations. The energy, computation, and link are the three broadly divided basic types of heterogeneity of WSN. Another vital factor to consider is the heterogeneity of data creation rates, which considers nodes with varying data transmission requirements. As a result, distinct performance evaluation parameters must be used to categorise sensor nodes. So there is a necessity to categorise sensor nodes based on different performance evaluation metrics. Motivated by the above facts, in this article, we provide a genetic algorithm-based energy-aware multi-hop clustering scheme for heterogeneous WSNs. Table 1 shows the Summarisation of Related Works.

The proposed method is similar to *Capone et al. (2019)* and *Pal et al. (2020)*. Compared to existing works (*Capone et al., 2019*; *Zhang et al., 2017*; *Darabkh, Zomot & Al-qudah, 2019*; *Javaid et al., 2013a*; *Sarkar & Senthil Murugan, 2019*; *Kumar & Kumar, 2016*; *Mann & Singh, 2017*; *Ali, Shahzad & Khan, 2012*; *Pal et al., 2020*), our study is distinguished by the type of algorithm. In this approach, two methods are investigated in HWSNs. The first method uses GA to enhance performance by selecting the optimal CHs during the clustering and re-clustering phases. The second method extends the first method by featuring optimal transmission scheduling. In this method, we carefully analyse the transmission scheduling and communication among CHs. As a result, we address various properties and analyses of node strategies to minimise the end-to-end delay, extend the network lifetime, and improve energy efficiency. However, this is the first article presenting a GA-based energy-aware multi-hop clustering to minimise end-to-end delay, expand the network lifetime, and enhance energy efficiency in HWSNs.

**Table 1 The summarisation of related work.**

| Name of the proposed solutions | Functionality | Advantages | Disadvantages |
|---|---|---|---|
| HEED (*Kumar et al., 2020*) | Cluster heads (CH) have been selected according to a hybrid of the node residual energy | Surely guarantee connectivity of clustered networks | It works only on two-level hierarchy, not to multilevel hierarchies |
| CATD (*Andrushia et al., 2021*) | It is improved in the cluster data transmission phase after the CHs are selected | It reduces the network energy, network overhead, and cost. | Hot-spot problems are created |
| EADC (*Garg et al., 2020a*) | It uses competition range to construct clusters of even sizes. | Achieves load balance among CHs | Uneven clustering strategy |
| ERA (*Chukwu, Garg & Zahra, 2021*) | The clever strategy of CH selection, residual energy of the CHs and the intra-cluster distance for cluster formation. | Achieves constant message and linear time complexity. | High message complexity for building backbone network of CHs. |
| S-MDSP (*Bugeja, Garg & Audu, 2016*) | Delay-minimization scheduling for multi-hop networks. | Minimising the end-to-end delay | The delay is significantly reduced by combining cooperative forwarding (CF) and forward interference cancellation (FIC). |
| E2HRC (*Shang, 2013*) | Messaging structure for clustering and routing | Balancing average energy consumption, network load and improving network performance | Delay occurred |
| EDB-CHS-BOF (*Amgoth & Jana, 2015*) | A tight closed-form expression for the optimal number of CHs in the network | Balancing energy consumption amongst all sensor nodes and prolonging the network lifetime. | It is more sensitive to any changes in the network size. |
| EDDEEC (*Jin et al., 2013*) | Probabilities for CH selection based on initial, the remaining energy level of the nodes and average energy of network | Achieves longer network lifetime and stability period | Dynamic random channel selection |
| FABC (*Saini & Sharma, 2010*) | optimally selecting cluster-head based on fitness function value of nodes | Maximise the network energy and lifetime of nodes | – |
| iABC (*Capone et al., 2019*) | Obtain optimal cluster heads | Improves energy efficiency in WSNs | – |
| MOPSO (*Pantazis, Nikolidakis & Vergados, 2013*) | optimise the number of clusters in an *ad hoc* network as well as energy dissipation in nodes | Provide an energy-efficient solution and reduce the network traffic | – |
| EEWC (*Afsar, Mohammad & Tayarani, 2014*) | Solve the clustering problem in a wireless sensor network | Improve the performance of WSNs | – |

# MATERIALS AND METHODS

In HWSNs, clusters are formed based on GA. GA finds optimal CHs by considering the network coverage and its energy level. The CHs perform data aggregation and transmit the combined data packets to the sink. A multi-hop network is used to send packets from CHs to the sink. Neighbouring sink nodes consider regular, advanced, and supernodes. These nodes have different initial energy. Regions near the sink have a more significant number of supernodes than other regions. The next-hop CH is selected with the distance between the CHs, the residual energy, the number of CMs, and the neighbouring CHs associated with the given CH in routing. Various symbols and notations used in the proposed work are mentioned in Table 2.

**Table 2 The list of symbols and notations in this article.**

| Symbol | Description |
| --- | --- |
| G | Bi-directed Graph |
| V | Network size |
| E | Two-way communication links |
| C | Direct links |
| $\gamma$ | Noise power |
| $\eta$ | SINR value |
| $\sigma$ | Energy at node |
| M | Data packet set |
| d | Distance between the nodes |
| $\rho$ | The energy dissipated in the source and sink |
| $\alpha, \beta$ | Space and multipath fading coefficients |
| $\varpi$ | Size of optimal CHs |
| $\vartheta$ | The distance among CHs and CMs |
| $\tau$ | Two CHs distance |

## Multi-hop network model

A WSNs is assumed to be a bidirected graph $G = (V, E, C)$, where $V$ denotes the network size, $E \subseteq V \times V$ is two-way communication links, and $C = \{(i,j), (j,i) : \{i,j\} \in E\}$ is the direct links. We consider that $\{i,j\} \in E$ iff, the SINR is convinced, i.e., $\frac{\sigma(i,j)}{\gamma} \geq \eta$ and $\frac{\sigma(j,i)}{\gamma} \geq \eta$, where $\sigma(i,j)$ and $\sigma(j,i)$ be the energy at node $j$ when node $i$ is sending and receiving data packets, respectively. It can be represented as $\sigma(i,j) := P(i)\varphi(i,j), \sigma(j,i) := P(j)\varphi(j,i)$, where $P(i)$ and $P(j)$ be the transmitted energy of nodes $i$ and $j$. Here, $\varphi(i,j) = \varphi(j,i)$ is to obtain the communication link $\{i,j\}$, $\gamma$ is the noise power and $\eta$ is the SINR value. The network size occurrence to $i \in V$ is defined by $\Gamma(i)$, i.e., $\Gamma(i) = \{j \in V : \{i,j\} \in E\}$.

Assume that the order of time $T := \{1, 2, ..., \tau\}$ and $\omega$ is the group of nodes in a time. The direct link $(i,j) \in C$ is very dynamic, only if $i \in \omega, j \notin \omega$ and the resulting SINR is convinced:

$$\frac{\sigma(i,j)}{\gamma + \sum_{\kappa \in \omega/\{i\}} \sigma(k,j)} \geq \eta \qquad (1)$$

A node can either send, receive, or be inactive at a particular time. A group $c \subseteq C$ of communication links will be simultaneously very active for the compatible set $(c)$ situations. The group of active sensor node $c$ is represented by $\omega(c) := \{i : \exists j, (i,j) \in c\}$. The SINR is applied to communication links $(i,j) \in c$:

$$\frac{\sigma(i,j)}{\gamma + \sum_{\kappa \in \omega(c)/\{i\}} \sigma(k,j)} \geq \eta \qquad (2)$$

Consider the data packet set M, and each data packet $m \in M$ needs a time for a particular transmission and is sent from source $S(m)$ to sink $D(m)$. The data packets are available in the respective sources on time $\tau = 1$. The time $\tau = \overline{T}$ occurs in transmitting the packets and computes the total network delay.

## Optimised energy model

The proposed GA-EMC is adopted an optimised energy model (*Abo-Zahhad et al., 2015*) that minimises energy consumption. The nodes' energy is needed to communicate a data packet consisting of $l$ bits of a packet is denoted by Eq. (3).

$$\sigma(l, d) = \begin{cases} l\rho + l\alpha d^2, & d < d_0 \\ l\rho + l\beta d^4, & d \geq d_0 \end{cases} \tag{3}$$

where, $d$ represents the distance between the nodes involved in the communication, $\rho$ is the energy dissipated in the source and sink. It considers the factors like modulation and digital coding. The variables $\alpha$ and $\beta$ represent space and multipath fading coefficients, and the threshold $d_0$ decide whether to use a multipath fading model.

Equation (4) gives the energy spent by the sink receiving $l$ bit of packets.

$$\sigma_R(l) = l \times \rho \tag{4}$$

The CM node spends the energy to send a packet to its CH. The power spent by the CM to transmit $l$ bits of a packet to its CH is determined by Eq. (5).

$$\sigma_{CM_i} = \sigma_T(l, d_{i,CH_i}) \tag{5}$$

where, $d_{i,CH_i}$ represents the Euclidean distance between the $i^{th}$ CM and its CH. A CH spends its power to receive a packet from its CMs, aggregate all the packets, and send it to other CHs. In addition to forwarding the local cluster data, CHs may also forward the traffic received from other CHs. Equation (6) shows the energy required by CHs.

$$\sigma_{CH_j} = N_j^{CM} \sigma_R(l) + N_j^{CM} l\sigma_A + \sigma_T(l, d_{j,P_j}) + \sigma_{CH_j}^F \tag{6}$$

where, $N_j^{CM} \sigma_R(l)$ denotes the energy spent by CH to accept data packets from the CMs. The second gives the energy spent in aggregation, the third term gives the energy spent for data transmission to the next-hop CH, and the last term $\sigma_{CH_j}^F$ represents the energy spent in forwarding the relay traffic. $\sigma_{CH_j}^F$ is the sum of energy required to receive $k$ bits of the packet from all the low-level CHs and communicate the packets to the parent CH as shown by Eq. (7),

$$\sigma_{CH_j}^F = N_{CH_j}^{CH}(\sigma_R(l) + \sigma_T(l, d_{j,P_j})) \tag{7}$$

## Phases in the proposed GA-EMC protocol

The proposed GA-EMC contains four phases: Heterogeneous Nodes Deployment, Clustering Formulation, Selection of Next-hop Neighbor, and Packet Transmission. The proposed GA-EMC scheme's main idea is to optimise energy management of the WSNs by minimising the intra-cluster distance between a CH and a CM. Using Euclidean distance,

the distance between the CM and the CH is calculated for WSNs. The CM is placed in the cluster with the least space between it and the others.

The nodes interact directly with the sink if the distance between the sink and the sensor node is smaller than the distance between CH and CM. When a node joins a cluster, it sends a JOIN message to the other nodes and the CH to let them know it's there. The CH assigns each node a time slot for data collection. After the data has been acquired, the CH aggregates it before sending it to the sink. The nodes may sleep during this entire process, but CH must be awake at all times. This lowers CH's energy, and few nodes die over time due to living in a sparse network. In each cycle, the clusters are reconstructed, and CHs are chosen.

The fitness function is used in the proposed GA-EMC technique to reduce the intra-cluster distance between the sensor nodes and the cluster head (CH). The function optimised the CH's placement, which impacts the estimated number of packet retransmissions along the path and hence on the network's overall energy usage. Because GA-EMC works with the fitness function, the proposed technique is preferable in terms of performance measurement in terms of energy consumption.

Minimising the distance between CMs and their CHs examines the sink distance, intra-cluster distance, and residual energy of CMs to determine their ideal positions. These phases are described below.

## Heterogeneous nodes deployment phase

In multi-hop communication, the CHs are situated very close to the sink node and have to forward more packets received from other nodes, and their power is exhausted quicker than the CHs far away from the sink. This creates a hot spot in the regions near the sink. To solve this issue, sensor nodes are classified into regular nodes with initial energy $\sigma_0$, advanced nodes with initial energy $\sigma_0(1 + \alpha)$, and supernodes with initial energy $\sigma_0(1 + \beta)$ joules. The value of energy heterogeneity constants $\alpha$ and $\beta$ are greater than 1. The WSNs consist of N nodes in total with $m_a \times$ N advanced nodes, $m_s \times$ N supernodes, and $(1 - m_a - m_s) \times$ N regular nodes. The areas near the sink node have more supernodes than the areas away from the sink.

## Clustering formulation phase

In this phase, more clusters are formed in HWSNs. It also contains two other sub-phases, namely CHs Selection and CM Association phases. The CHs selection phase selects an optimal CH. Each CM is associated with any one of the nearest energy-efficient CH in the CM association phase.

### CH selection phase

This phase uses the GA for selecting optimal CHs and their location. GA is working on natural genetics and natural selection principles and is used to optimise various parameters. GA is applied in multiple fields for solving constrained and unconstrained optimisation problems (*Bandyopadhyay & Coyle, 2003*).

### CM association phase

Each CH sends a CH advertisement ($CH_{ADV}$) message containing its identifier, location, initial and residual energies and starts a clustering timer with a predefined value in the CM association phase. When a CM receives $CH_{ADV}$, it stores the information in the cluster table (CT). A CM may receive $CH_{ADV}$ from one or more CHs. For each CH entry stored in the CT, CM calculates the cost as given by Eq. (8). CM selects their respective CH with low cost and sends JOIN message to the optimal CH.

$$Cost = c_1 \times f_1 + (1 - c_1) \times f_2 \tag{8}$$

Here $c_1$ is a constant $0 \le c_1 \le 1$. By setting proper value for $c_1$, we can decide how much importance to give to distance and energy in the CH selection. The terms $f_1$ and $f_2$ are calculated as provided by Eqs. (9) and (10).

$$f_1 = \frac{d(CM_i, CH_i^j)}{d_{\max}} \tag{9}$$

$$f_2 = \frac{\lambda_i^j}{\varphi_i^j} \tag{10}$$

where, $\lambda_i^j$ and $\varphi_i^j$ represents the initial and residual energies of $CH_i^j$ respectively. $CH_i^j$ is a CH present in the cluster table of $CM_i$ and $d_{\max}$ is the distance among the CH and its CMs, and it is calculated by Eq. (11),

$$d_{\max} = \max \left\{ d(CM_i, CH_i^j) \right\} \tag{11}$$

The CHs collect the JOIN message from the CMs until the clustering timer expires. Upon the expiry of the timer, CHs create a dynamic time division multiple access (TDMA) scheduling for the packet transmission and send it to the CMs. The GA-based clustering algorithm shows the various steps involved in forming clusters and optimal CH selection.

### Next-hop neighbour selection phase

Each CH broadcasts a neighbour advertisement message that contains information like identifier, location, initial and residual energies, distance to sink, and the size of CMs associated with it. When a CH receives a neighbour advertisement message, it adds the information contained in the packet to the neighbour. As shown in Fig. 1, CHs use multi-hop paths to communicate the data packets to the sink. The next-hop CHs are chosen based on the distance, residual energy, and the size of CMs associated with the next-hop CH, the number of CHs that have reached *via* the next-hop CH. When more CHs can be reached *via* a CH, the CH will help forward the packet reliably. CH with more residual energy, less distance, smaller CMs, and more neighbouring CHs prefer the next-hop CH.

For each CH node in the neighbour table, a merit value (MV) is calculated based on the above factors. Equation (12) shows the calculation of MV.

$$MV = \theta_1 \times \delta_1 + \theta_2 \times \delta_2 + \theta_3 \times \delta_3 + \theta_4 \times \delta_4 \tag{12}$$
$$\theta_1 + \theta_2 + \theta_3 + \theta_4 = 1 \tag{13}$$

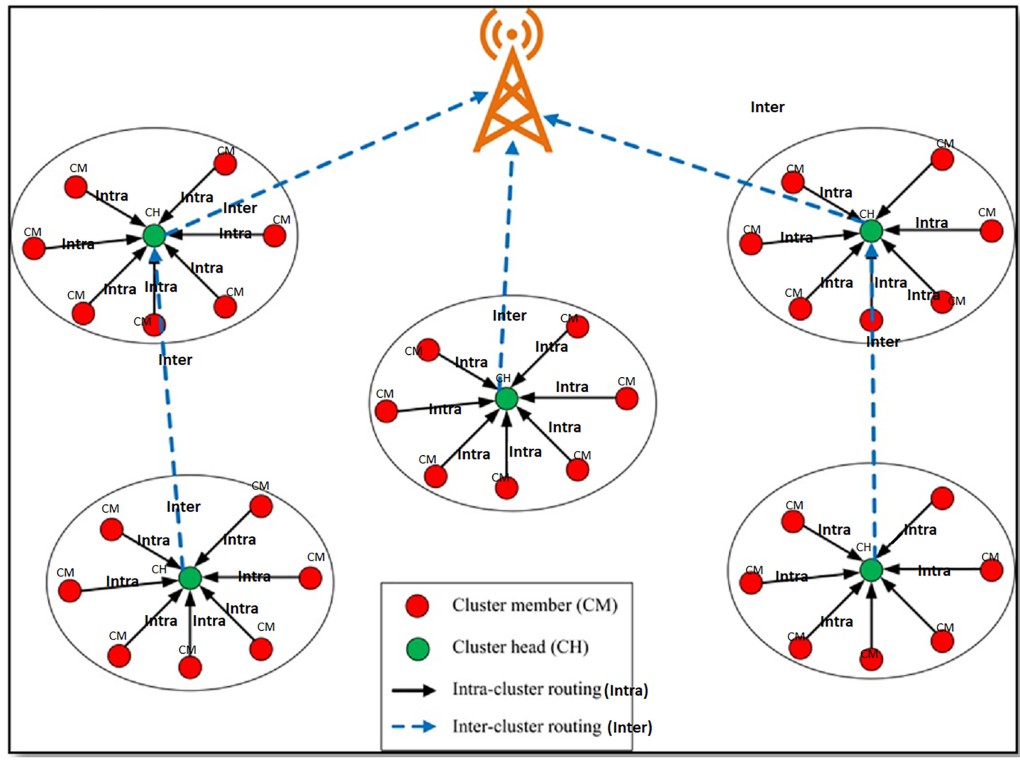

**Figure 1 Multi-hop communication from CH to sink.** The red dots represent the cluster member (CM). The green dots represent the cluster head (CH). Solid line arrows represent the intra-cluster routing. Dotted lines represent the inter-cluster routing.

$$\delta_1 = \frac{(d(CH_i, CH_j) + (CH_j, S)}{d_{\max_i}} \tag{14}$$

$$d_{\max_i} = \max \{d(CH_i, CH_j) + (CH_j, S)\}, \ j \in \theta \tag{15}$$

Here $\theta$ represents the neighbouring CHs of $CH_i$.

$$\delta_2 = \frac{\rho_j}{\varphi_j} \tag{16}$$

$$\delta_3 = N_{CH_j}^{CM}, \ j \in \theta \tag{17}$$

$$\delta_4 = \frac{1}{N_{CH_j}^{CM}} \ j \in \theta \tag{18}$$

In Eq. (13), $\theta_1, \theta_2, \theta_3,$ and $\theta_4$ represents weights associated with different factors.

### Data transmission phase

It involves communication within the cluster and communication between sink and CH. In intra-cluster communication, the CH receives packets from their CMs per the dynamic TDMA scheduling. CM also senses the data from the surroundings and sends them to the concerned CHs during a particular time. The CMs turn off their radio in the remaining time to save the energy wasted during idle listening. Each CH has many next-hop CH neighbours, and the best neighbour node is selected in the next-hop neighbour selection phase.

## Genetic algorithm

In GA, each result to a specific problem is denoted by a chromosome using a binary coding scheme. A group of chromosomes constitutes the population. The initial population consists of randomly selected chromosomes, and each bit in the chromosome is called a gene. For each chromosome, a fitness value is calculated, and it evaluates the effectiveness of the chromosome. Chromosomes with high fitness values will get more chances to create new chromosomes. The GA involves three basic operations: selection, crossover, and mutation to select the best chromosome. The selection process duplicates good chromosomes and eliminates the poor ones, and there are many selection methods like tournament selection, ranking selection, and roulette wheel selection. The crossover operation selects two parents, recombines them, and creates two children. Crossover can be either single-point crossover or multi-point crossover. Crossover does not introduce any new genetic properties. Mutation operation introduces new genetic properties. These operations are repeated for a given number of generations (*Bandyopadhyay & Coyle, 2003*; *Pantazis, Nikolidakis & Vergados, 2013*). The implementation of various GA operations is explained below.

*i. Binary Coding*: Binary coding scheme represents each chromosome for the given sensor scenario as a string of $0_s$ and $1_s$. a chromosome of length N bits signifies HWSNs with N nodes. The chromosome size is the same as the size of the network. In the chromosome set, value 1 and 0 represents the CH and CM, respectively. Figure 2 shows the chromosome representation of a network with 20 sensor nodes. Nodes $S_2, S_3, S_{11}$ are CHs, and the remaining nodes are CMs.

*ii. Objective Function*: The objective function ($\delta$) is used for selecting optimal CHs. In designing $\delta$, the following facts are considered. The optimal CH consumes more energy than the CM, so the number of CHs must be minimised. The power required for intra-cluster communication depends on the distance between CHs and CMs, and the power required for inter-cluster communication depends on the distance between two CHs. To save power, we have to reduce the size of optimal CHs ($\varpi$), the distance between CHs and CMs ($\vartheta$), and the two CHs distance ($\tau$). By selecting CHs with higher residual energy, we can deliver packets reliably. The $\delta$ selects the CHs by considering the above factors, and it is a minimisation function as given in Eq. (19).

$$\delta = \left(\frac{1}{\varphi}\right) + \vartheta + \tau + \varpi \tag{19}$$

where, $\varphi$ represents the sum of the residual energy associated with the CHs.

$$\varphi = \sum_{i=1}^{\varpi} \varphi_i \tag{20}$$

Eq. (21) determines the sum of the distance between CMs from their respective CHs.

$$\vartheta = \sum_{i=1}^{\varpi} \sum_{i \in C_i} d(CM_i^j, CH_i) \tag{21}$$

| S1 | S2 | S3 | S4 | S5 | S6 | S7 | S8 | S9 | S10 | S11 | S12 | S13 | S14 | S15 | S16 | S17 | S18 | S19 | S20 |
|----|----|----|----|----|----|----|----|----|-----|-----|-----|-----|-----|-----|-----|-----|-----|-----|-----|
| 0 | 1 | 1 | 0 | 0 | 0 | 0 | 0 | 0 | 0 | 1 | 0 | 0 | 0 | 0 | 0 | 0 | 0 | 0 | 0 |

**Figure 2 Binary coding representation of a chromosome.** The first row represents sensor nodes and the second row represent their corresponding binary coding. The chromosome size is the same as the size of the network. In the chromosome set, value 1 and 0 represents the CH and CM, respectively.

where, $CH_i$ represents the $i^{th}$ CH, $C_i$ denotes the set of CMs associated with $CH_i$ and $CM_j$ represents $j^{th}$ CM node associated with $CH_i$.

In Eq. (22), $\tau$ represents the total distance between all the CHs in the $i^{th}$ level to the parent CH nodes in the $i-1^{th}$ level. The node level is considered to find out the parent CH nodes. All the CHs in level 1 send packets to the destination directly. CHs in the remaining level send packets to their parent CHs in a multi-hop fashion to the sink, and CHs in $i-1^{th}$ level is the parent of CHs in the $i^{th}$ level.

$$\tau = \sum_{i=1,j=i-1}^{\varpi} d(CH_i, CH_j) \tag{22}$$

where $j \in P_i$ and $P_i$ represents the set of parent CHs associated with $CH_i$.

***iii. Fitness Function:*** GA is generally suitable for solving the maximisation problem. Since our aim is minimising $\delta$, this problem is transformed into maximising the fitness value $f_v$. For each chromosome in the population $\delta$ is used to calculate the $f_v$ as given by Eq. (23).

$$f_v = \frac{1}{1+\delta} \tag{23}$$

***iv. Selection:*** It is used to select chromosomes with higher $f_v$ to join the mating pool to form a new population for the subsequent generations. The proposed method uses the Roulette wheel selection method.

***v. Crossover:*** The proposed GA-EMC scheme uses a single-point crossover. A random value (0 to 1) and two chromosomes have been selected for this operation. The crossover operation is performed only if the selected random value is less than the crossover probability $p_c$. Otherwise, no crossover is done. If it is decided to perform crossover, an arbitrary crossover point is selected. After the crossover point, the two-parent chromosomes exchange their packet to generate two child chromosomes. Table 3 shows the crossover operation.

***vi. Mutation:*** In bit-level mutation, a random value is chosen for every bit in a chromosome. Suppose this random value is less than $P_m$, then the mutation is performed to invert the bit. Otherwise, the bit is kept as such.

As shown in Table 4, in the first chromosome, no mutation is performed, whereas in the second chromosome, six bits are mutated.

Selection, mutation, and crossover operations are repeated for given generations. The better chromosome is selected at the end of the last generation. In the best

chromosome selection, if the genome value is 1, the node becomes CH, and otherwise, it becomes CM.

## GA-based clustering algorithm

Begin
    Choose binary coding to represent chromosome
    Set values for population size
    Set values for cross over and mutation probability
    Set the maximum number of generations $g_{max}$
    Initialise generation counter $g_0$
    Generate initial population
while ($g < g_{max}$) do
    Fitness function used for evaluating chromosomes in the population
    Perform reproduction
    Perform crossover
    Perform mutation
    Increment the generation counter by 1
end while
    Get the best chromosome
for $i = 1$ to $N$
if (chromosome $[i] = = 1$) then
    Get the position of the CH nodes
for each CH do
    send $CH_{ADV}$ message
    while the clustering timer has not expired
        Receive JOIN message
end while
create and send TDMA schedule to CMs
end for
else
    While the clustering timer has not expired
        Receive $CH_{ADV}$ message
    add information about CHs in the CT
    end while
for every record in the CT do
    Find cost
    JOIN the CH with minimum cost
    receive TDMA schedule from the CH and
        transmit data according to it
end for
end if
end for
end

**Table 3  A single-point crossover.**

| Parent chromosome | Cross over point | Child chromosome |
| --- | --- | --- |
| 00010001110000000101 | 16 | 00010001110000001101 |
| 00001001010000101101 | 16 | 00001001010000100101 |

**Table 4  Bit-level mutation.**

| Chromosome before mutation | Chromosome after mutation |
| --- | --- |
| 00010001110000001101 | 00010001110000001101 |
| 00001001010000100101 | 00001100010010100010 |

## Minimising end-to-end delay with packet forwarding mechanism

Let $\tau$ represent an upper limit on the delay with $\mathrm{T} = \{1, 2, ..., \tau\}$. A mathematical model is designed to analyse the number of data packets sent and received between CH and CMs for a particular time. We use the binary variables: $\psi^t = 1$ if time $t \in \mathrm{T}$; $Z_{i,s}^t = 1$ if node $i \in v$ is transmitting $s \in S$ in $t \in \mathrm{T}$; $Y_{i,s}^t = 1$ if node $i \in v$ is receiving $s \in S$ in $t \in \mathrm{T}$; $\kappa_{i,s}^t = 1$ if $s \in S$ is present at $i \in v$ by $t \in \mathrm{T}$.

GA-EMC is specially formulated to minimise the delay required to send packets from source to destination. The constraint $\psi^{t+1} \leq \psi^t, t \in T \backslash \{\tau\}$ has been forced all the time after the first round. The constraint $\sum_{s \in S} \left( Z_{i,s}^t + Y_{i,s}^t \right) \leq \psi^t, i \in v, t \in T$ ensures that a node can either send and receive data packets at a particular time or nothing to be done. The constraint $\sum_{i \in v} Z_{i,s}^t \leq 1, \sum_{i \in v} Y_{i,s}^t \leq 1, s \in S, t \in \mathrm{T}$ ensures that, in time $t$, the data is transmitted by one node and received by another node in HWSNs. The constraint $\sum_{i \in v(j)} Z_{i,s}^t \geq Y_{j,s}^t, j \in V, s \in S, t \in \mathrm{T}$ ensures that the node $j$ receives a packet $s$ in the current time. Inequality $\sum_{i \in T} Z_{i,s}^t \leq 1, \sum_{t \in T} Y_{i,s}^t \leq 1, i \in V, s \in S$ allows a node to communicate data packets during the time. The constraint $\sum_{i \in v} Z_{i,s}^t \leq 1, \sum_{i \in v} Y_{i,s}^t \leq 1, s \in S, t \in \mathrm{T}$ is fully justified in transmitting packets through multi-hop routing. The constraints $a_{i,s}^t \leq \sum_{T=1}^{t} Y_{i,s}^T, s \in S, i \in V \backslash \{O(s)\}, t \in T$ and $a_{O(s)s}^1 = 1, a_{D(s)s}^T = 1, s \in S$ defines variable $a$ and set the conditions for the starting and ending of the dynamic TDMA scheduling. Finally, the constraint $Z_{i,s}^t \leq a_{i,s}^t, i \in V, s \in S, t \in T$ expresses the SINR state for sending data packets $s$ on link $(i, j)$ at a time $t$. Subsequently, the SINR state is stable when $Z_{i,s}^t = Y_{j,s}^t = 1-$ agreeing to the case when all nodes besides only node $i$ is sending packets in a network. Although, node $j$ receives a data packets $s$ from node $i$ in a time $t$, then $\sum_{t \in \mathrm{T}} \psi^t$ becomes equivalent to

**Table 5 The parameters and values for simulation.**

| Parameters | Value |
|---|---|
| Network area | $800 \times 800 \ m^2$ |
| Network size | 400 |
| The initial energy of the normal node | 0.5 *Joule* |
| Packet size | 4 *KB* |
| $\eta$ | 10 *pJ/bit/m²* |
| $\gamma$ | 0.0013 *pJ/bit* |
| $\rho$ | 50 *nJ/bit* |
| $\alpha$ and $\beta$ | 1 and 2 |
| $P_m$ | 0.6 |
| $p_c$ | 0.03 |
| Population size | 20 |
| Maximum generations | 20 |

$$p(i,j) \geq \partial(\beta + \sum_{i \in V \setminus \{i,j\}} \sum_{s \in S \setminus \{s\}} p(k,j)Z_{ks}^t), \tag{24}$$

which accurately confirms that the SINR value is met. In Eq. (24), $\sum_{s \in S \setminus \{s\}} p(k,j)Z_{ks}^t)$, is valid since when $Z_{i,s}^t = 1$ then all the nodes besides $i$ are illegal to send $s$ in $t$ since

$$\sum_{i \in v} Z_{i,s}^t \leq 1, \sum_{i \in v} Y_{i,s}^t \leq 1, s \in S, t \in \text{T}.$$

We observe that the packet forwarding mechanism is used to increase the transmissions in HWSNs. In GA-EMC, the packet forwarding mechanism increases the transmissions at a particular time, and more data packets are transmitted to the CMs through adjacent clusters. This is possible for increasing the use of packet forwarding and forward interference cancellation mechanisms among CMs in all clusters in the ensuing time, which is more cooperative for minimising delay in HWSNs.

# RESULTS

In this section, the performance of GA-EMC is analysed and compared with E-MDSP (*Capone et al., 2019*) and EEWC (*Pal et al., 2020*). Simulations are performed using the Network Simulator (NS2) (*Issariyakul & Hossain, 2012*). An HWSN consists of 400 nodes in a simulation area. To evaluate the GA-EMC performance, we have considered the metrics such as network lifetime, throughput, network stability, the number of data packets sent to the sink, and the average energy consumption in the whole network. The various parameters for simulation are presented in Table 5.

## Network lifetime

To extend the HWSNs' lifetime, we have considered the alive nodes in each round. Figure 3 illustrates that the proposed GA-EMC scheme extends the lifetime of alive nodes in every round than EEWC and E-MDSP. The proposed GA-EMC provides a better network lifetime than existing schemes.
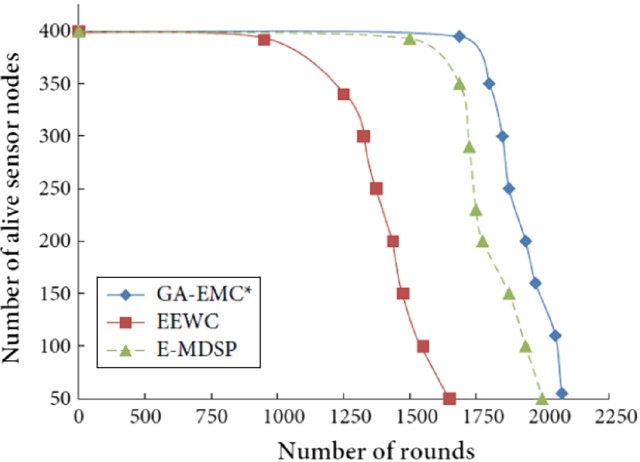

**Figure 3 Number of rounds *vs* number of alive nodes.** The X-axis represents the number of rounds, and Y-axis represents the number of alive sensor nodes. Green-line, red-line and blue-line represent proposed GA-EMC, E-MDSP and EEWC, respectively.

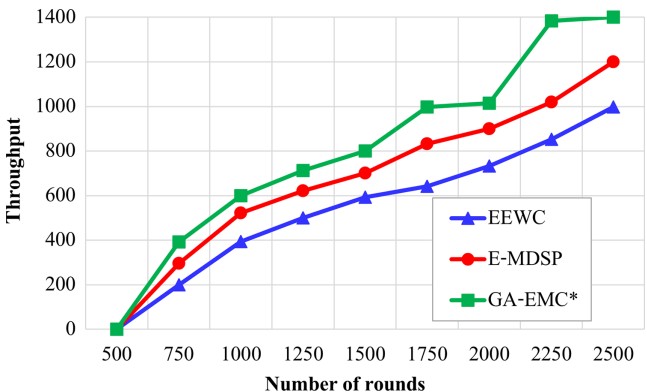

**Figure 4 Number of rounds *vs* throughput.** The X-axis represents the number of rounds and Y-axis represents throughput. Green-line, red-line and blue-line represent proposed GA-EMC, E-MDSP and EEWC respectively. The EEWC performs poorly with less data packet communication. Similarly, the E-MDSP gives the best behaviour than EEWC and also provides poor performance than GA-EMC.

The proposed GA-EMC uses multi-hop communication for packet delivery to extend the network lifetime. Compared to the existing schemes, the first node dies after 1,800 rounds in GA-EMC. Later, the last node remains alive for 2,100 rounds. In EEWC and E-MDSP schemes, the nodes have died after 1,000 and 1,600 rounds, respectively. Figure 3 shows that the proposed GA-EMC scheme prolongs the network lifetime and stability, and the last alive node can still respond to the network in this approach.

## Throughput

In HWSNs, the proposed GA-EMC algorithm analysed the number of data packets sent, each CH sends data packets to the sink, and the CMs send data packets to their respective CHs. As shown in Fig. 4, the EEWC performs poorly with less data packet communication. Similarly, the E-MDSP gives the best behaviour than EEWC and also provides poor

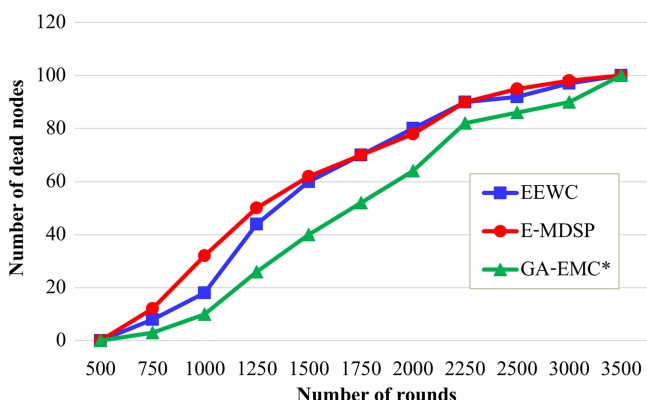

**Figure 5 Number of rounds *vs* number of dead nodes.** The X-axis represents the number of rounds, and Y-axis represents the number of dead nodes. Green-line, red-line and blue-line represent proposed GA-EMC, E-MDSP and EEWC, respectively. It shows the regular time interval from the beginning of the network process until the death of the first node in HWSNs. The GA-EMC has a better stability period than the other schemes. The first dead node starts at 1,800 rounds in the GA-EMC scheme, whereas the first dead node starts nearly 1,000 and 1,600 rounds under the EEWC, E-MDSP approaches. The stability duration of GA-EMC compared with the EEWC scheme increases from 1,000 to 2,500 rounds, and the E-MDSP increases from 1,600 to 2,500 rounds. So, GA-EMC provides better stability duration and prolongs the network lifetime.

performance than GA-EMC. The number of data packets sent from CHs is increased significantly by the EMC-GA and achieves better throughput when compared to the other schemes.

## Stability period

Figure 5 illustrates the regular time interval from the beginning of the network process until the death of the first node in HWSNs. As shown in Fig. 5, the GA-EMC has a better stability period than the other schemes. The first dead node starts at 1,800 rounds in the GA-EMC scheme, whereas the first dead node starts at nearly 1,000 and 1,600 rounds under the EEWC, E-MDSP approaches. The stability duration of GA-EMC compared with the EEWC scheme increases from 1,000 to 2,500 rounds, and the E-MDSP increases from 1,600 to 2,500 rounds. So, GA-EMC provides better stability duration and prolongs the network lifetime.

## Minimising the end-to-end delay

Figure 6 displays the analysis of various approaches in terms of delay. It shows that the EEWC acquires the extreme delay of 0.04 s in 2,500 rounds. However, the delay is low compared to EEWC, and it is maximum than the GA-EMC. At the same time, E-MDSP achieves a minimum delay than EEWC, but it fails to outperform GA-EMC. GA-EMC approach achieves a low delay of only 0.02 s at the 2,500 rounds.

## The average energy consumption

Figure 7 displays the average energy consumption under variable simulation rounds.
Even though more packets are transmitted in the proposed protocol than in EEWC and E-MDSP, the average energy consumption till a particular round is less in the proposed

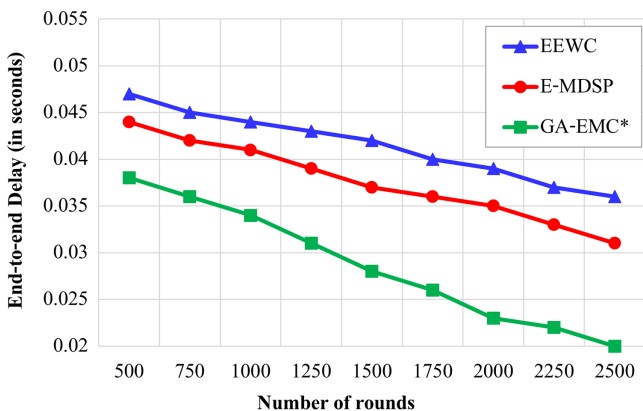

**Figure 6 Number of rounds *vs* delay.** The X-axis represents the number of rounds, and Y-axis represents the End-to-end Delay (in seconds). Green-line, red-line and blue-line represent proposed GA-EMC, E-MDSP and EEWC, respectively. The EEWC acquires the extreme delay of 0.04 s in 2,500 rounds. However, the delay is low when compared to EEWC, and it is maximum than the GA-EMC. At the same time, E-MDSP achieves a minimum delay than EEWC, but it fails to outperform GA-EMC. GA-EMC approach achieves a low delay of only 0.02 s at the 2,500 rounds.

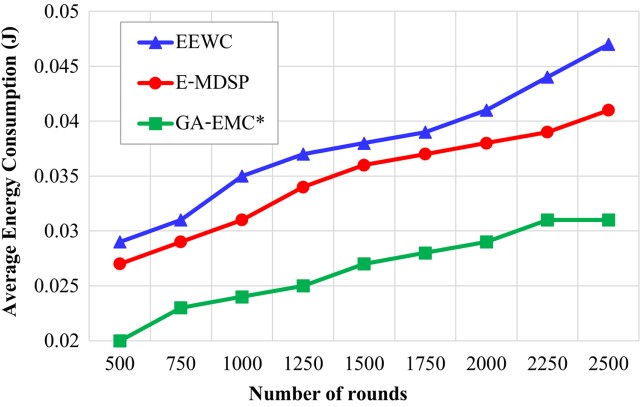

**Figure 7 Number of rounds *vs* average energy consumption.** The X-axis represents the number of rounds, and Y-axis represents the average energy consumption under variable simulation rounds. Green-line, red-line and blue-line represent proposed GA-EMC, E-MDSP and EEWC, respectively. Even though more packets are transmitted in the proposed protocol than EEWC and E-MDSP, the average energy consumption till a particular round is less in the proposed GA-EMC.

GA-EMC, as shown in Figure 7. This energy-saving aims to use multi-hop communication and associate the CMs with the optimum CHs.

## Impact of sink node location on the HWSNs lifetime and stability

Network stability is measured by the round when the first node died. To study the impact of sink location on the network stability and lifetime, we have considered three scenarios. In scenarios 1, 2, and 3, the sink is situated at various places such as the middle, top right corner, and outside the field, respectively. Table 6 showed the comparison of the round when a given percentage of nodes died for different sink positions. On average, the

**Table 6 Comparison of percentage of nodes died for different sink node locations.**

| % of nodes died | Number of rounds | | | | | | | | |
|---|---|---|---|---|---|---|---|---|---|
| | Sink located at center 100, 100 | | | Sink located at top right corner 200, 200 | | | Sink located outside the field 100, 300 | | |
| | GA-EMC | E-MDSP | EEWC | GA-EMC | E-MDSP | EEWC | GA-EMC | E-MDSP | EEWC |
| 10 | 1,049 | 1,019 | 717 | 710 | 680 | 401 | 312 | 282 | 123 |
| 20 | 1,192 | 11,62 | 922 | 824 | 794 | 564 | 534 | 504 | 256 |
| 30 | 1,412 | 1,382 | 1,028 | 978 | 948 | 621 | 678 | 648 | 384 |
| 40 | 1,554 | 1,524 | 1,087 | 1,126 | 1,096 | 789 | 756 | 726 | 490 |
| 50 | 1,693 | 1,663 | 1,157 | 1,267 | 1,237 | 854 | 869 | 839 | 587 |
| 60 | 1,765 | 1,735 | 1,245 | 1,398 | 1,368 | 981 | 1,023 | 993 | 712 |
| 70 | 1,958 | 1,928 | 1,353 | 1,501 | 1,471 | 1,102 | 1,265 | 1,235 | 891 |
| 80 | 2,196 | 2,166 | 1,521 | 1,689 | 1,659 | 1,256 | 1,412 | 1,382 | 1,038 |
| 90 | 2,458 | 2,428 | 1,702 | 1,876 | 1,846 | 1,411 | 1,623 | 1,593 | 1,214 |
| 100 | 2,641 | 2,611 | 1,813 | 2,123 | 2,093 | 1,714 | 1,831 | 1,801 | 1,485 |

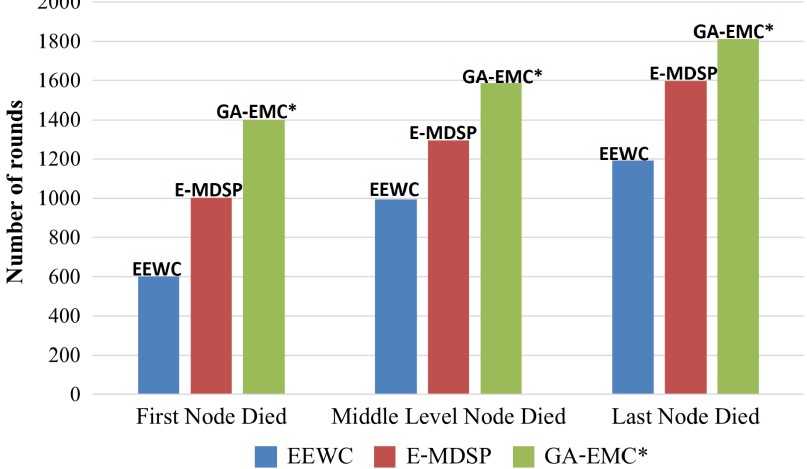

**Figure 8 Comparing GA-EMC with EEWC and E-MDSP based on network lifetime.** The X-axis represents each of three scenarios: *i.e.*, the first node died, the middle node died and the last node died. The Y-axis represents the rounds when the first node died in the three scenarios. Green-bar, red-bars and blue-bars represent proposed GA-EMC, E-MDSP and EEWC, respectively.

proposed protocol extends the round when the last node died by 30.94% by considering different sink positions. GA-EMC extends the network lifetime and better stability in all three cases, and it provides more significant improvement when the sink is at the corner and outside the field due to multi-hop routing.

Figure 8 shows the rounds when the first node died in the three scenarios. As shown in Fig. 8, the round when the first node died is postponed by 10.98%, 23.47% and 46.94% in scenarios 1, 2 and 3, respectively. This shows that the proposed protocol performs better for longer distance transmission. Compared to EEWC and E-MDSP, the GA-EMC

provides a 27.13% improvement in the round when the first node died, considering the average of different sink positions.

## DISCUSSION

The proposed GA-EMC scheme outperforms the existing methods, especially EEWC and E-MDSP, in almost all aspects. It extends the lifetime of alive nodes in every round and prolongs the network lifetime and stability. Also, it significantly increases the number of data packets sent from CHs and achieves better throughput. It provides better stability duration and prolongs the network lifetime.

Furthermore, it achieves a lower delay and reduces the average energy consumption till a particular round. It extends the network lifetime and better stability in all three cases. Due to multi-hop routing, it improves when the sink is at the corner and outside the field and performs better for longer distance transmission.

## CONCLUSIONS

In this article, a GA-EMC scheme is presented for extending the lifetime and minimising the delay in HWSNs. In selecting the optimal CHs, the fitness value is calculated based on cluster distances, the number of CHs, and their initial and residual energies. Each cluster selects a CH with minimum distance, higher residual energy, minimum CMs, and maximum neighbours as its next hop in inter-cluster routing. The energy hole problem created due to multipath routing is solved by deploying more higher energy supernodes in the areas closer to the sink. The mathematical model for energy consumption for clustering with multi-hop data transmission is explained. The experimental results proclaim that GA-EMC prolongs the HWSNs lifetime, minimises the delay, and maximises stability compared to EEWC and E-MDSP for various positions of the BS, primarily when the BS is situated in the network corner and outer area. The death of the first and last nodes is prolonged by 27.13% and 30.94%, respectively, compared with EEWC and E-MDSP. In the future, the simulation can be repeated to see the impact of the number of nodes in HWSNs. Also, the performance of GA_EMC can be analysed in actual (not simulated) HWSNs in some practical scenarios.

## ACKNOWLEDGEMENTS

We thank reviewers, editors and publishers for providing valuable feedback to improve the manuscript. We also appreciate Vija Prakash for helping us with formatting.

### Funding

The work was funded by NICE-Healthcare, Research Excellence Funds by The University of Malta. There was no additional external funding received for this study. The funders had no role in study design, data collection and analysis, decision to publish, or preparation of the manuscript.

## Grant Disclosures
The following grant information was disclosed by the authors:
NICE-Healthcare.
Research Excellence Funds by the University of Malta.

## Competing Interests
Lalit Garg & NZ Jhanjhi are Academic Editors for PeerJ.

## Author Contributions
- R. Muthukkumar conceived and designed the experiments, performed the experiments, prepared figures and/or tables, authored or reviewed drafts of the article, and approved the final draft.
- Lalit Garg analyzed the data, prepared figures and/or tables, authored or reviewed drafts of the article, and approved the final draft.
- K. Maharajan performed the experiments, analyzed the data, performed the computation work, prepared figures and/or tables, and approved the final draft.
- M. Jayalakshmi performed the experiments, analyzed the data, performed the computation work, prepared figures and/or tables, and approved the final draft.
- Nz Jhanjhi performed the computation work, prepared figures and/or tables, authored or reviewed drafts of the article, and approved the final draft.
- S. Parthiban analyzed the data, prepared figures and/or tables, authored or reviewed drafts of the article, and approved the final draft.
- G. Saritha analyzed the data, prepared figures and/or tables, authored or reviewed drafts of the article, and approved the final draft.

## Data Availability
   The code is available in the Supplemental File.

## Supplemental Information
Supplemental information for this article can be found online at http://dx.doi.org/10.7717/peerj-cs.1029#supplemental-information.

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
