# Peer review of "A genetic algorithm-based energy-aware multi-hop clustering scheme for heterogeneous wireless sensor networks"

_PeerJ Computer Science, doi:10.7717/peerj-cs.1029_

## Round 0.1 · original submission · Major Revisions

This is an interesting work, but there is scope to improve as per the reviews done. Please work on all the reviewer comments with justifications where needed and address all the points raised, to improve the clarity and quality of the paper.

The reviewers have requested that you cite specific references. You may add them if you believe they are especially relevant. However, I do not expect you to include these citations, and if you do not include them, this will not influence my decision.

Thank you.

Reviewer 1 ·

Basic reporting

This paper proposes a genetic algorithm-based energyaware multi-hop clustering (GA-EMC) scheme for heterogeneous WSNs (HWSNs). In HWSNs, all the nodes have varying initial energy and typically have an energy consumption restriction. A genetic algorithm determines the optimal CHs and their positions in the network. The fitness of chromosomes is calculated in terms of distance, optimal CHs, and the node's residual energy. Multi-hop communication improves energy efficiency in HWSNs. The areas near the sink are deployed with more supernodes far away from the sink to solve the hot spot problem in WSNs near the sink node. Results. Simulation results proclaim that the GA-EMC scheme achieves a more extended network lifetime, network stability and minimises delay than existing approaches in heterogeneous nature.
Detailed comments are as follow:
(1) MAC protocol has an important impact on network performance, and it is also an important research content of clustering networks. Therefore, I strongly suggest that the author discuss the recent MAC protocol in the relevant work. For example: A parallel joint optimized relay selection protocol for wake-up radio enabled WSNs," Physical Communication, vol. 47, 101320, august 2021.
(2) Some problems have been addressed by the authors, the reviewer strongly suggests that the theoretical analysis of the system performance should be added to improve the quality of this paper.The author can found such work in:"heoretical analysis of the lifetime and energy hole in cluster based Wireless Sensor Networks[J],Journal of Parallel and Distributed Computing, 2011,71(10):1327-1355."
(3) Authors are suggested to review more new and relevant research to support their research contribution.Many references in this paper are the work of more than 10 years ago.
(4) I suggest that the author set up real experiments to test the performance of the prposed protocol.
(5) The formulas in the text is incorrectly formatted,which should be align with the words instead of being upper than the words.
(6) In fact, the work of this paper has been studied in many previous works. I also believe that the work done by the author is effective. However, the innovation of the paper is not strong, and almost all the work done in the paper can be found in the previous work. However, the author gives a PSO algorithm in detail, which can be used as a reference for relevant work. Therefore, it is suggested to give the author a chance to modify it to improve the quality of the paper.

Experimental design

Please see the "Basic reporting".

Validity of the findings

Please see the "Basic reporting".

Additional comments

Please see the "Basic reporting".

Reviewer 2 ·

Basic reporting

In this paper, a genetic algorithm-based energy-aware multi-hop clustering (GA-EMC) scheme for heterogeneous WSNs (HWSNs) is proposed. Genetic algorithm determines the optimal CHs and their positions in the network. The fitness of chromosomes is calculated in terms of distance, optimal CHs, and the node's residual energy. Multi-hop communication improves energy
efficiency in HWSNs. Simulation results proclaim that the GA-EMC scheme achieves a more extended network lifetime, network stability and minimises delay than existing approaches in heterogeneous nature. This work is much meaningful, my comments are as follows:
1) Energy-efficient hierarchical clustering on IoT/WSNs is a very well-studied area with a lot of previous studies and over saturated. Therefore, there needs to be a very strong motivation and justification for the proposed approach not to repeat the previous contributions.
2) Given that there are tons of works in WSNs research, the related work should be written in such a way that the reader can see the differences of works in a Table and grasp the main assumptions/disadvantages of other approaches. In literature review, it is very important to summarize the advantages and disadvantages of existing work.
3) This work uses GA to optimize the energy consumption in WSNs, which have been deeply studied so far. Hence, the analysis in the paper on WSNs with GA is necessary, the following article is also appropriate choices:
[1] X.-Y. Zhang, J. Zhang, Y.-J. Gong, Z.-H. Zhan, W.-N. Chen, and Y. Li, “Kuhn–Munkres parallel genetic algorithm for the set cover problem
and its application to large-scale wireless sensor networks,” IEEE Trans. Evol. Comput., vol. 20, no. 5, pp. 695–710, Oct. 2016.
[2] Y. Chang, X. Yuan, B. Li, D. Niyato, N. Al-Dhahir, "A Joint Unsupervised Learning and Genetic Algorithm Approach for Topology Control in Energy-Efficient Ultra-Dense Wireless Sensor Networks", Communications Letters IEEE, vol. 22, no. 11, pp. 2370-2373, 2018.
[3] Y. Chang, X. Yuan, B. Li, D. Niyato, and N. Al-Dhahir, Machinelearning-based parallel genetic algorithms for multi-objective optimization in ultra-reliable low-latency WSNs, IEEE Access, vol. 7, pp. 4913–4926, 2019.
4) There can be the situation that some CHs are overloaded as they may have many nearby CMs. It might be good to state how this issue can be avoided. In addition, how to formulate the fitness in the optimization process?

Experimental design

The simulation workload is enough, but the comparison algorithm needs to be added.

Validity of the findings

Need enhancement.

Reviewer 3 ·

Basic reporting

Overall the paper is well-presented and clearly written in professional, unambiguous language. The topic has much significance. The novelty of the work and the contributions are clearly reported. I commend the authors for their technical correctness, extensive experimentation, and analysis. The performance of the proposed work has been compared with the state-of-the-art approaches which establish the validity of the proposed protocol. However, the authors are suggested to take into account the following issues for further improvement:
1. The introduction contains mostly older references. Hence, the authors are suggested to include a few recent research in this section.
2. The authors could present the existing research regarding multi-hop clustering schemes in WSNs according to the timeline so that a thorough gap analysis could be made.
3. The authors could list the frequently used symbols with their meaning in a table for convenience.
4. The authors are suggested to go through the entire manuscript thoroughly to correct grammatical mistakes.

Experimental design

1. What is the complexity of the proposed algorithm to execute it at runtime?
2. As mentioned in the text, the implementation of the proposed protocol would require a few additional control packet exchanges such as CH advertisement, JOIN message, control packets for next-hop CH selection, etc. It would be appreciable if a comprehensive analysis of control packet overhead for executing the proposed protocol at runtime is presented.
3. It would be better if the input and the output of the proposed GA-based Clustering Algorithm are also mentioned in the algorithm presentations
4. The authors could provide further details regarding experimental setup such as the transmission power used, the initial energy of the advanced nodes and supernodes, the radio model used, the transceiver used, etc.
5. What is the mode of operation of the proposed cluster selection technique? Does every node or any dedicated node will execute the GA-based Clustering Algorithm at runtime?

Validity of the findings

1. The authors could illustrate why the proposed protocol outperforms the existing works while explaining the findings of the experiments.
2. The sink positions could be included in Figure 8 for a better understanding of the findings. The authors may also give a separate plot for analyzing the performance of the proposed protocol for different sink positions.

Annotated reviews are not available for download in order to protect the identity of reviewers who chose to remain anonymous.

---

## Round 0.2 · Minor Revisions

If the reviewers have given references to add, please ignore them. Please make sure the paper is complete in every way, based on the remaining review suggestions.

Reviewer 1 ·

Basic reporting

The authors have addressed my concern and it can be accepted now.

Experimental design

The authors have addressed my concern and it can be accepted now.

Validity of the findings

The authors have addressed my concern and it can be accepted now.

Additional comments

The authors have addressed my concern and it can be accepted now.

Reviewer 2 ·

Basic reporting

This manuscript is good writing.

Experimental design

Good.

Validity of the findings

The proposed algorithm is enough innovative.

Additional comments

I agree the responds from authors.

·

Basic reporting

This revised paper is much better now. But it needs major revision due to the following reasons.
1. Please strictly follow this journal template, like alignment, section number etc.

2. Please cite reference papers with increasing references order.

3. In the related works part, it is not suggested to mention each reference paper with 1-2 setences, which is not meaningful.
It is suggested to firstly classfy them into several types and then give explanation about their own work (uniqueness).

4. All tables and figures are missiong in the main pdf file.

5. All the symbols are not alligned well.

6. Please change "Where" to "where" and move it to the front on certain line below certain equations.

7. Acknowledgements part is missing.

8. Reference part is weak, and ref. 93 is missing. There are too many so-so references, which is not necessary.
And more relevant papers about " energy efficiency and optimization for WSN" is suggested like below.
--A PSO based Energy Efficient Coverage Control Algorithm for Wireless Sensor Networks, Computers Materials & Continua,vol.56, no.3, pp.433-446, 2018.
--Optimal Coverage Multi-Path Scheduling Scheme with Multiple Mobile Sinks for WSNs, Computers, Materials & Continua, vol.62, no.2, 2020, pp.695-711.
-- An Enhanced PEGASIS Algorithm with Mobile Sink Support for Wireless Sensor Networks, Wireless Communications & Mobile Computing, Volume 2018, Article ID 9472075, 2018.
--Multiple Strategies Differential Privacy on Sparse Tensor Factorization for Network Traffic Analysis in 5G, IEEE Transactions on Industrial Informatics, vol.18, no.3, pp.1939-1948, 2022.
--A novel fault tolerance energy-aware clustering method via social spider optimization (SSO) and fuzzy logic and mobile sink in wireless sensor networks (WSNs), Computer Systems Science and Engineering, vol. 35, no.6, pp. 477–494, 2020.
--Global levy flight of cuckoo search with particle swarm optimization for effective cluster head selection in wireless sensor network, Intelligent Automation & Soft Computing, vol. 26, no.2, pp. 303–311, 2020.

Experimental design

This revised paper is much better now. But it needs major revision due to the following reasons.
1. Please strictly follow this journal template, like alignment, section number etc.

2. Please cite reference papers with increasing references order.

3. In the related works part, it is not suggested to mention each reference paper with 1-2 setences, which is not meaningful.
It is suggested to firstly classfy them into several types and then give explanation about their own work (uniqueness).

4. All tables and figures are missiong in the main pdf file.

5. All the symbols are not alligned well.

6. Please change "Where" to "where" and move it to the front on certain line below certain equations.

7. Acknowledgements part is missing.

8. Reference part is weak, and ref. 93 is missing. There are too many so-so references, which is not necessary.
And more relevant papers about " energy efficiency and optimization for WSN" is suggested like below.
--A PSO based Energy Efficient Coverage Control Algorithm for Wireless Sensor Networks, Computers Materials & Continua,vol.56, no.3, pp.433-446, 2018.
--Optimal Coverage Multi-Path Scheduling Scheme with Multiple Mobile Sinks for WSNs, Computers, Materials & Continua, vol.62, no.2, 2020, pp.695-711.
-- An Enhanced PEGASIS Algorithm with Mobile Sink Support for Wireless Sensor Networks, Wireless Communications & Mobile Computing, Volume 2018, Article ID 9472075, 2018.
--Multiple Strategies Differential Privacy on Sparse Tensor Factorization for Network Traffic Analysis in 5G, IEEE Transactions on Industrial Informatics, vol.18, no.3, pp.1939-1948, 2022.
--A novel fault tolerance energy-aware clustering method via social spider optimization (SSO) and fuzzy logic and mobile sink in wireless sensor networks (WSNs), Computer Systems Science and Engineering, vol. 35, no.6, pp. 477–494, 2020.
--Global levy flight of cuckoo search with particle swarm optimization for effective cluster head selection in wireless sensor network, Intelligent Automation & Soft Computing, vol. 26, no.2, pp. 303–311, 2020.

Validity of the findings

This revised paper is much better now. But it needs major revision due to the following reasons.
1. Please strictly follow this journal template, like alignment, section number etc.

2. Please cite reference papers with increasing references order.

3. In the related works part, it is not suggested to mention each reference paper with 1-2 setences, which is not meaningful.
It is suggested to firstly classfy them into several types and then give explanation about their own work (uniqueness).

4. All tables and figures are missiong in the main pdf file.

5. All the symbols are not alligned well.

6. Please change "Where" to "where" and move it to the front on certain line below certain equations.

7. Acknowledgements part is missing.

8. Reference part is weak, and ref. 93 is missing. There are too many so-so references, which is not necessary.
And more relevant papers about " energy efficiency and optimization for WSN" is suggested like below.
--A PSO based Energy Efficient Coverage Control Algorithm for Wireless Sensor Networks, Computers Materials & Continua,vol.56, no.3, pp.433-446, 2018.
--Optimal Coverage Multi-Path Scheduling Scheme with Multiple Mobile Sinks for WSNs, Computers, Materials & Continua, vol.62, no.2, 2020, pp.695-711.
-- An Enhanced PEGASIS Algorithm with Mobile Sink Support for Wireless Sensor Networks, Wireless Communications & Mobile Computing, Volume 2018, Article ID 9472075, 2018.
--Multiple Strategies Differential Privacy on Sparse Tensor Factorization for Network Traffic Analysis in 5G, IEEE Transactions on Industrial Informatics, vol.18, no.3, pp.1939-1948, 2022.
--A novel fault tolerance energy-aware clustering method via social spider optimization (SSO) and fuzzy logic and mobile sink in wireless sensor networks (WSNs), Computer Systems Science and Engineering, vol. 35, no.6, pp. 477–494, 2020.
--Global levy flight of cuckoo search with particle swarm optimization for effective cluster head selection in wireless sensor network, Intelligent Automation & Soft Computing, vol. 26, no.2, pp. 303–311, 2020.

---

## Round 0.3 · accepted · Accept

The authors have addressed all the concerns raised by the reviewers.